# Developing New Treatment Options for Castration-Resistant Prostate Cancer and Recurrent Disease

**DOI:** 10.3390/biomedicines10081872

**Published:** 2022-08-03

**Authors:** Bo-Ren Wang, Yu-An Chen, Wei-Hsiang Kao, Chih-Ho Lai, Ho Lin, Jer-Tsong Hsieh

**Affiliations:** 1Division of Urology, Department of Surgery, Taichung Armed Forces General Hospital, Taichung 41152, Taiwan; bozn123@gmail.com; 2Department of Urology, University of Texas Southwestern Medical Center, 5323 Harry Hines Blvd., Dallas, TX 75390, USA; yachen.cmu@gmail.com (Y.-A.C.); kao790108@gmail.com (W.-H.K.); 3National Defense Medical Center, Taipei 11490, Taiwan; 4Department of Life Sciences, National Chung Hsing University, Taichung 40227, Taiwan; hlin@dragon.nchu.edu.tw; 5Department of Microbiology and Immunology, Graduate Institute of Biomedical Sciences, College of Medicine, Chang Gung University, Taoyuan 33302, Taiwan; chlai@mail.cgu.edu.tw

**Keywords:** castration-resistant prostate cancer, recurrent therapy and castration-resistant prostate cancer, precision medicine

## Abstract

Prostate cancer (PCa) is a major diagnosed cancer among men globally, and about 20% of patients develop metastatic prostate cancer (mPCa) in the initial diagnosis. PCa is a typical androgen-dependent disease; thus, hormonal therapy is commonly used as a standard care for mPCa by inhibiting androgen receptor (AR) activities, or androgen metabolism. Inevitably, almost all PCa will acquire resistance and become castration-resistant PCa (CRPC) that is associated with AR gene mutations or amplification, the presence of AR variants, loss of AR expression toward neuroendocrine phenotype, or other hormonal receptors. Treating CRPC poses a great challenge to clinicians. Research efforts in the last decade have come up with several new anti-androgen agents to prolong overall survival of CRPC patients. In addition, many potential targeting agents have been at the stage of being able to translate many preclinical discoveries into clinical practices. At this juncture, it is important to highlight the emerging strategies including small-molecule inhibitors to AR variants, DNA repair enzymes, cell survival pathway, neuroendocrine differentiation pathway, radiotherapy, CRPC-specific theranostics and immune therapy that are underway or have recently been completed.

## 1. Introduction

Prostate cancer (PCa) is the most common cancer and the second deadliest cancer among men in the United States, which is mainly due to metastatic disease [1]. In general, surgery or radiation is potentially a curative treatment for localized disease. Since PCa is characterized as a typical androgen-dependent disease [2], hormone therapy (i.e., androgen deprivation therapy (ADT)) is the most effective therapy to control metastatic disease. However, almost all patients eventually develop castration-resistant PCa (CRPC) within 12 to 18 months, with a median survival of 14 to 26 months [3]. Nowadays, new anti-androgens (Enzalutamide or Abiraterone), radiotherapy (Radium-223) or immunotherapy (sipuleucel-T) have been approved for metastatic CRPC (mCRPC) patients to prolong the overall survival. Inevitably, mCRPC further acquires resistance and becomes therapy- and castration-resistant PCa (t-CRPC), which is considered as an end-stage disease [4,5] without effective therapy, and on which new therapeutic strategies have been actively explored.

Clinical observation indicates that about 20–25% of CRPC patients recurred from ADT [6,7], harbored distinct cell features from classical adenocarcinoma in prostate; some carcinoma cells exhibit neuroendocrine phenotypes with neuronal markers expression and neuronal factors secretion in an endocrine fashion [8], which is considered as neuroendocrine PCa (NEPC), a subpopulation of t-CRPC. However, small-cell carcinoma of prostate (SCPC) with very low incidence (1% of the prostatic malignancies) has been identified from the primary site. Although SCPC is associated with a highly proliferative area of tumor mass and poor prognosis, it is still sensitive to chemotherapy [9,10]. On the other hand, NEPC is known to resist many therapeutic regimens. Currently, no effective targeted therapy for NEPC has been approved by the FDA. Based on molecular profiling from NEPC patients, this article has discussed several potential new therapeutic strategies for this disease.

## 2. Hormone Therapy

### 2.1. Enzalutamide

In PCa, androgen receptor (AR) activated by androgens still represents a critical oncogenic pathway. Enzalutamide is a novel antiandrogen agent that can block AR with high affinity compared to traditional antiandrogens such as bicalutamide or flutamide [11]. Besides direct binding to AR, it can reduce AR translocation into the nucleus and prevent its transcription by binding to DNA [12] (Figure 1).

Enzalutamide was approved by the Food and Drug Administration (FDA) in 2012 for use on metastatic CRPC based on the randomized, phase III trial study (AFFIRM) [13,14] (Table 1). The study demonstrated that patients who received enzalutamide after chemotherapy improved median survival from 13.6 to 18.4 months (HR: 0.63 *p* < 0.001). Another phase III, double-blind, randomized study (PREVAIL) (Table 1) compared patients receiving enzalutamide before chemotherapy and showed that radiographic progression-free survival (rPFS) and overall survival (OS) benefit from 31.3 to 35.3 months [15,16]. It also can delay the initiation of chemotherapy in patients with metastatic PCa. In the phase III randomized trial (PROSPER), enzalutamide demonstrated the benefit of metastasis-free survival rate compared to placebo (36.6 months vs. 14.7 months, *p* < 0.001) in patients with non-metastatic CRPC (nmCRPC) [17] (Table 1). Therefore, the FDA expand the use of enzalutamide in patients with nmCRPC since 2018.

In patients with metastatic, hormone-sensitive PCa (mHSPC), enzalutamide improved the overall survival rate and progression-free survival over standard care in the phase III, randomized trial (ENZAMET) [18] (Table 1). In another double-blind, randomized trial (ARCHES), enzalutamide also revealed the benefit of rPFS (not-reached vs. 19 months, *p* < 0.001) in mHSPC patients [19] (Table 1). Regarding the safety and adverse effects of the drug, enzalutamide had the adverse effect to the central nervous system via entering the blood–brain barrier. Therefore, seizure and even posterior encephalopathy syndrome were reported in rare cases [12,17].

### 2.2. Apalutamide

Apalutamide is a second-generation androgen inhibitor and is already approved for use in nmCRPC and mHSPC. A double-blind, placebo-controlled, phase III trial (SPARTAN) demonstrated that apalutamide improved median metastasis-free survival compared with placebo (40.5 vs. 16.2 months. *p* < 0.001) in patients with non-mCRPC [20] (Table 1). Time to symptomatic progression was significantly longer with apalutamide than with placebo [20]. Another randomized phase III trial (TITAN), in the patients with mHSPC, apalutamide plus ADT, significantly improved OS and rPFS compared with ADT plus placebo [21] (Table 1).

The adverse effects of interest include fracture, dizziness, and hypothyroidism. Compared with other androgen inhibitors with enzalutamide and darolutamide, skin rash was mostly found in apalutamide [12,20,21].

### 2.3. Darolutamide

Like other second-generation androgen inhibitors, darolutamide can inhibit AR translocation, DNA binding, and AR-mediated transcription (Figure 1). From the phase I/II study, darolutamide inhibited cell proliferation more efficiently than enzalutamide in a castration animal model [33]. Besides, it also blocks the activity of the mutant ARs like the F876L mutation caused by enzalutamide or apalutamide [33].

In a randomized, double-blind, placebo-controlled, phase III trial (ARAMIS), darolutamide improved metastasis-free survival (40.4 months vs. 18.4 months, *p* < 0.001) compared with placebo in patients with nmCRPC [22] (Table 1). Additionally, it also demonstrated benefit in OS, time to pain progression, time to cytotoxic chemotherapy, and time to a symptomatic skeletal event [22]. Another randomized, phase III trial (ARASENS), combination therapy with darolutamide, ADT, and docetaxel improve OS over placebo plus ADT and docetaxel in patients with mHSPC [34].

Generally, darolutamide causes less toxic effect because of low blood–brain barrier penetration and low binding affinity for γ-aminobutyric acid type A receptors due to its unique structure, which is different from enzalutamide and apalutamide. Therefore, it causes less central nervous system effect like a seizure [12,22]. The most reported adverse effects were fatigue and asthenic conditions [22].

### 2.4. Abiraterone

Abiraterone acetate is a selective CYP17 enzyme inhibitor that can decrease the synthesis of androgen of the testis, adrenal gland and prostate gland [35,36,37,38] (Figure 1). In the double-blind, placebo-controlled phase 3 study (CO-AA-301 clinical trial), it demonstrated that abiraterone combined with prednisolone improved OS compared to the placebo plus prednisolone group (15.8 vs. 11.2 months. *p* < 0.0001) in patients with mCRPC who progressed after chemotherapy [23] (Table 1). Another phase III study (CO-AA-302) further confirmed the benefit of abiraterone used in patients with mCPRC before chemotherapy. It improved rPFS and OS (34.7 months vs. 30.3 months) compared with placebo [24] (Table 1). The FDA approved the use of abiraterone in patients with mCRPC post-chemotherapy in 2011, subsequently, pre-chemotherapy based on these two phases III clinical trials in 2012. Furthermore, the FDA further approved the use of abiraterone in patients with high-risk mHSPC based on the phase III clinical trial (LATITUDE study) in 2018 [25,26] (Table 1). Abiraterone plus ADT and prednisolone improved OS and rPFS in patients with mHSPC compared with placebo plus ADT, because the use of abiraterone may cause the miner alocorticoid increase, which will cause fluid edema, liver function impairment, low potassium level, and high blood pressure [23,24,25]. Therefore, abiraterone must be used in combination with steroid, and regular follow-up of liver function and potassium level is necessary [23,24,25]. Previous study indicated that the E3 ubiquitin ligase adaptor speckle-type pox virus and zinc finger (POZ) protein (SPOP) are involved in controlling protein stability of AR by interacting with steroid receptor coactivator 3 (SRC-3) protein and some of the transcriptional coactivators [39]. Wild-type SPOP can promote the SRC-3 protein and then suppress AR transcriptional activity, thus functioning as a potential tumor suppressor [39]. However, some studies identified mutant SPOP in 6–15% of PCa and it can alleviate the tumor suppressive effect [40,41]. Surprisingly, PCa patients with mutant SPOP appear to delay the onset of ADT resistance to 42.0 (95% CI: 25.7–60.8) months. Also, the better outcomes in patients with mutant SPOP treated with abiraterone and enzalutamide were observed [42,43]. Therefore, it implies that SPOP mutation could enhance ADT effect by maintaining AR dependency. Inflammatory response is related to patients treated with ADT. It can cause a IL-6 increase through testosterone suppression [44] and disease progression to CRPC through the activation STAT3 pathway in PCa [45], because IL-6 is known to induce epithelial–mesenchymal transition (EMT) leading to PCa invasion [46]. It is warranted that inflammation factors can be of prognostic value in PCa progression [45].

## 3. AR Splice Variant-7 (AR-V7) Inhibitors

For mCRPC patients, drug resistance to 2nd-generation AR signaling inhibitors (ARSi), such as abiraterone and enzalutamide is essentially universal in tumor cells that often come with significantly elevated expression of truncated AR splice variant-7 (AR-V7) [47]. Although enzalutamide had a significant effect on CRPC, about 20–40% of patients had no response to the drug. A study showed that AR-V7 was associated with resistance to abiraterone and enzalutamide [48]. Therefore, developing a novel technique for detecting AR-V7 in circulating cells is helpful for the treatment of patients with CRPC [49]. While the expression of AR-V7 was rare in primary PCa (<1%) but common in mCRPC (75%), AR-V7 protein is expressed after primary ADT alone in those CRPC patients, and further increases during 2nd-generation ARSi therapy [50]. Therefore, there is an urgently needed new treatment to reduce the impact of the elevated AR-V7 expression, leading to lethal progression of CRPC. Niclosamide is an anthelminthic drug approved by the FDA; it can decrease the protein expression of AR-V7 in CRPC cells through the ubiquitin–proteasome pathway [51]. Two in vitro studies indicated that niclosamide combined with abiraterone or enzalutamide significantly decreased tumor volume compared to abiraterone or enzalutamide alone, suggesting that niclosamide treatment can restore sensitivity to 2nd-generation ARSi therapy [51,52]. In addition, two types of clinical research trials assess the combination treatment with enzalutamide (Phase I, NCT03123978) or abiraterone (Phase II, NCT02807805) with niclosamide for CRPC patients.

HSP90 is a molecular chaperon essential for the transcriptional activity of the androgen receptor and enhance stability, function, and regulation in the late stage [53,54]. Ferraldeschi et al. demonstrated that the second-generation HSP90 inhibitor, onalespib, can reduce AR-V7 mRNA levels, and not total AR transcript levels [55]. It may be beneficial for patients with PCa expressing AR-V7 and support further research in HSP90 inhibitors in patients resistant to ARSi.

## 4. PROTAC (Proteolysis Targeting Chimera) Degrader

The development of PROTEC is a new strategy for cancer therapy. PROTEC is composed of a POI (protein of interest) ligand, E3 ubiquitin ligase ligand, and a linker [56,57]. The heterobifunctional molecule can form a complex between the POI and E3 ligase. Then, the E3 will transfer ubiquitin to the surface of the POI. Finally, the proteasome will identify the polyubiquitination signal and degrade the POI [58,59]. Therefore, many AR PROTEC degraders like ARV-110 [58,59,60], ARD-61 [61], ARV-766 (phase I clinical trial, NCT05067140) [60], AR-LDD [60], ARCC-4 [62] are studied if depletion of AR can overcome the drug resistance of PCa. Another estrogen receptor (ER) PROTEC degrader, ARV-471, is also under clinical study for localized advanced or metastatic breast cancer [58] (Phase I/II clinical trial, NCT04072952). Many mutated genes in AR are identified in circulating cells of advanced PCa patients, including L702H0, T878A, H875Y, W742C, W742L, F877L, and T878S [63]. These findings encourage us to further understand the relationship between AR gene mutation and drug resistance. ARV-110 (bavdegalutamide) is a novel, oral PROTAC degrader targeting wild-type and relevant mutant ARs. According to the data of a phase I/II clinical trial reported in the 2022 GU ASCO (American Society of Clinical Oncology Genitourinary Cancers Symposium, San Francisco), ARV-110 lowers 50% of PSA levels in nearly half of patients with AR T878X (T878X = T878A or T878S) or H875Y mutations [60] who have been validated to be drug targets. PSA decline is also observed in patients without AR T878X or H875Y mutations.

## 5. Radiotherapy

### 5.1. Bone Target Agent-RADIUM-223 (Ra-223)

Over 90% of patients with mCRPC will eventually develop bony metastatic diseases [27,64], which usually cause mortality, movement disability, and complications. Traditional bone targeting agents like bisphosphonates and denosumab do not reveal benefits in overall survival and only relieve pain and skeletal events. Ra-223 is an a-particle emitter to target a high turnover area of bone [27] by causing DNA breaks, leading to a bystander effect of adjacent tumor suppression [65,66], in which it minimizes the effect to the surrounding healthy tissues, especially bone marrow [66].

A phase III clinical trial (ALSYMPCA) demonstrated that Ra-223 revealed OS benefit in mCRPC patients with symptomatic bony metastasis (14.9 vs. 11.3 months, *p* < 0.001) [27]. Ra-223 can provide a potent treatment for CRPC patients with bony metastases and elicit less adverse effects, particularly, myelosuppression [27,67] (Table 1). The occurrence rate of grade 3–4 myelosuppression was low, but much higher in a previous docetaxel group compared with a no chemotherapy group [67].

### 5.2. ^177^Lu-PSMA

Prostate-specific membrane antigen (PSMA) is a type II transmembrane glycoprotein expressed in the prostate epithelium [68]. It may perform multiple physiological functions including cell migration, nutrition uptake, and signal transduction [68]. PSMA expression increases from benign prostate epithelium to high-grade prostatic intraepithelial neoplasia or adenocarcinoma [69]. Besides, PSMA expression seems to be inversely related to the androgen level [70,71]. The PSMA activity increases in PCa cells lines as cells become more androgen-independent [71], which is different from PSA [71]. The PSMA receptor has the internalization process that can cause endocytosis in the putative ligand into the cell, which allows PSMA-labelled radioisotope to be more concentrated within the cell [72,73]. Due to the above characteristic, it is helpful to develop novel therapeutic methods to target the delivery of drugs, short-range radioisotope, and toxin specifically for mCRPC. The value of PSMA in position-emission tomography-computed tomography (PET/CT) is confirmed by a prospective study of 314 patients. Caroli et al. demonstrated that, in patients with biochemical recurrence of PCa, 68Ga-PSMA PET/CT showed superior performance and safety compared to choline PET/CT [74]. Ideally, the therapeutic radionuclides should match the lesion size/volume to be treated so that the energy can focus on the lesion, rather than the surrounding tissues [73]. ^177^Lu is a medium-energy b-particle emitter with a maximum energy of 0.5 MeV and relatively short particle tissue range of about 1.5 mm. The shorter b-range of ^177^Lu provides better irradiation to a relatively small tumor, compared with longer b-range of ^90^Y [73]. It also has a relatively long physical half-life of 6.73 days [73]. Therefore, these physical activities make ^177^Lu-PSMA an ideal therapeutic radionuclide that allows the delivery of high energy to target PCa cells. Anti-PSMA radionuclide therapies consist of monoclonal antibodies targeting the extracellular domain of PSMA (^177^Lu-PSMA-J591) or radiolabeled small molecules targeting the glutamate carboxypeptidase II pocket of PSMA (^177^Lu-PSMA- 617) [75]. Several studies reported that ^177^Lu-PSMA-J591, a monoclonal antibody to PSMA, has effective results showing 11.4% to 59.6% of PSA decline [75,76,77,78,79]. However, it also has higher myelosuppression rates and limited treatment response rates compared with studies using PSMA-labelled smaller molecules (^177^Lu PSMA-617) [80,81]. In addition, in a two-arm phase II study (TheraP study), 200 mCRPC patients were enrolled to compare the antitumor effect between ^177^Lu-PSMA-617 and cabazitaxel, which showed the better PSA response rates and fewer adverse effects with ^177^Lu-PSMA-617 monotherapy [82]. Subsequently, a phase III study (VISION study) was designed to explore the clinical benefit of ^177^Lu-PSMA-617 in mCRPC patients with progressive disease under hormone or chemo-therapy, and the results indicated that ^177^Lu-PSMA-617 can prolong rPFS and OS of advanced status of mCRPC patients with PSMA-positive expression [28] (Table 1). Notably, the most common adverse events associated with the ^177^Lu-PSMA-617, such as fatigue, dry mouth, were mild (grade I-II) [28], which promotes ^177^Lu-PSMA-617 as a new radiotherapeutic for t-CRPC patients. Furthermore, more clinical trials (phase III, NCT04720157 and phase III, NCT04689828) are underway to explore the clinical applicability of ^177^Lu-PSMA-617 in the early stages of PCa.

## 6. Immune Therapy

### 6.1. GM-CSF

Granulocytic-macrophage colony-stimulating factor (GM-CSF) is used as an adjuvant therapy in immunotherapy and has the potential to enhance antitumor efficacy [29]. Sipuleucel-T is the first immunotherapy approved by the FDA [83] (Table 1), which is an autologous dendritic cell therapeutic vaccine designed to enhance the immune T-cell response. It is prepared from peripheral blood mononuclear cells obtained by leukapheresis. These cells are exposed ex vivo to a novel recombinant protein immunogen with prostatic acid phosphatase and human GM-CSF [29]. Nevertheless, the immunologic effects of GF-CSF are not fully understood. Wei et al. demonstrated that GM-CSF can recruit effector T cells into the tumor microenvironment in localized PCa [84]. However, in the phase III trial, PROSTVAC, a viral vector-based immunotherapy which prolonged median overall survival (OS) in mCRPC [85,86], failed to confirm benefit in OS when in combination with GF-CSF [87].

### 6.2. PD1 Inhibitor (Pembrolizumab)

Programmed death ligand-1 (PD-L1) is expressed in many human tissues including tumor cells and can bind to PD-1, an immune checkpoint inhibitor expressed on T cells. Through the mechanism, tumor cells can escape the immune toxicity. Therefore, PD-1 inhibitors and PD-L1 inhibitors are studied to kill cancer cells via breaking the immune resistance [88,89] (Figure 1).

In a study, it demonstrated that, in treatment with PD-1 inhibition with nivolumab across tumor types including melanoma, non-small-cell lung cancer (NSCL), renal cancer (RCC), CRPC, and colorectal cancer [89], PD-1 inhibitors showed objective responses in approximately one in four to one in five patients with NSCL, melanoma, or RCC [89]. However, no objective responses (i.e., radiographic responses in soft-tissue) were observed in patients with CRPC [88,89].

The five cohorts, open label, phase II trial, KEYNOTE-199 study demonstrated that the objective response rate of pembrolizumab monotherapy was only 5% (95% CI, 2% to 11%) in patients with CRPC who had positive PD-1 expression and 3% (95% CI, <1% to 11%) in those with negative results. Disease control rates was 10% in those with positive PD-1 expression and 9% in those with negative PD-1 expression [90].

Many studies indicated that the prostate tumor microenvironment is highly immunosuppressive [88,91]. One study demonstrated that tumor-infiltrating lymphocytes (TILs) may contribute to PCa progression by inhibiting the activity of T-effector cells [92].

Another study indicated that adenosine produced by prostatic acid phosphatase and TGF-β can contribute to immunosuppression [91]. Therefore, although there are some immunotherapy trials in PCa, they resulted in limited results because of the immunosuppressive mechanism in PCa.

Nevertheless, the FDA approved the use of pembrolizumab for solid tumors expressing mismatch repair deficient (dMMR) or microsatellite instability-high (MSI-H) since 2017 [38,93]. The indication included several cancer types, not PCa specifically. It was based on five MSI-H studies involving treatment with pembrolizumab for colorectal cancers (*n* = 90) or non-colorectal cancers (*n* = 49) [93]. Among the non-colorectal cancer patients, two were mCRPC. The total objective response was 40% (59/149) [93]. Of the two patients with mCRPC, one achieved partial response, and the other achieved stable condition over nine months [93]. The prevalence of MSI-high(H)/deficient MMR (dMMR) has been observed in approximately 2–3% of cases in PCa among studies [94,95,96]. Mismatch repair (MMR) and microsatellite instability (MSI) are important mechanisms related to cancer progression. Many MMR enzymes (MLH1, MSH2, MSH6 and PMS2) are related to MSI when deficiency is present [97,98]. MMR is a mechanism which can replace and repair the wrong mismatches in daughter DNA strands [99]. On the other hand, the mechanism of microsatellite generation, consisting of repeated sequences of 1–6 nucleotides, can cause DNA slippage in the process of replication, or mismatch of the basic group of slippage strand, resulting in more repeating units missing [97,99].

A retrospective multi-institutional case series study demonstrated that patients with advanced microsatellite instability-high (MSI-H) prostate adenocarcinoma identified with clinical cell-free DNA (cfDNA) sequence testing treated with PD1 inhibitor (pembrolizumab) after two lines of therapy [97]. Of the total 14 participants, 9 CRPC patients have 56% bone, 33% nodal, 11% soft tissues, and 11% liver metastasis and median PSA level 29.3 ng/dL. PSA 50 was defined as PSA level declined ≥50% from baseline prior to treatment. After a median of 9.9 months of treatment, four (44%) in nine patients with mCRPC achieved PSA50 after a median of 4 weeks after treatment initiation, including three patients with >99% PSA decline [97].

The FDA further approved the use of pembrolizumab in patients with unresectable or metastatic tissue tumor mutational burden (tTMB)-high (≥10) tumors that have progressed with previous treatment without satisfactory outcomes since 2020. It was based on a multi-cohort, open-label, non-randomized, phase II KEYNOTE-158 study. It evaluated patients treated with the anti-PD-1 monoclonal antibody pembrolizumab in patients with selected, previously treated, advanced solid tumors [100]. It included unresected or metastatic tumors, 6 of whom had PCa [101]. Objective responses were observed in 30 (29%; 95% CI 21–39) of 102 patients in the tTMB-high group and 43 (6%; 5–8) of 688 in the non-tTMB-high group [100,101].

In addition, some studies demonstrated that mutant SPOP can stabilize programmed death ligand 1 (PD-L1) protein expression via proteasome-mediated degradation and improve PCa treatment outcome on programmed death 1 (PD-1) immune checkpoint inhibitors [102,103].

## 7. Chimeric Antigen Receptor (CAR) T-Cells

The recent development of biological engineering techniques has made a significant improvement in overcoming the immune-evasiveness of cancer cells. Clinically, CAR-T cell therapy is the first to be applied on several hematological malignancies [104], which is believed to be the same target antigens applicable for other somatic cancers [105]. However, the physical barriers such as tumor microenvironment of solid tumors probably account for the difficulties in obtaining the same promising results [105]. Nevertheless, PSMA and PSCA are the most important candidates for CAR-T cells targeted antigens in CRPC [105]. The process begins from T-cell collection through leukapheresis. T cells are genetically altered to express CAR which can target tumor-associated antigens (TAA) [106]. Lymphodepletion is usually performed before injection, which can cause CAR-T cells lymphopenia and more suitable to the tumor microenvironment [107]. Finally, purified CAR-T cells are administrated back to patients [106] (Figure 1). Different strategies are developed to create CAR-T cells. Comparing with autologous CAR-T cells, allogeneic CAR-T cells appear to be less expensive, more efficient from a well-established standardized protocol [108]. PSMA-targeted CAR-T cell is superior to other TAA CAR-T therapies for more easily monitored by the ^68^Ga-PET/CT examination [109,110,111].

Until now, two anti-PSMA CAR-T trials have been reported. In the clinical trial (NCT00664196) of the first generation of anti-PSMA CAR-T-cells therapy, PSA decline in 50% and 70% was found in two patients, but three other patients experienced disease progression [110]. The other clinical trial (NCT01140373) of the second-generation anti-PSMA CAR-T-cells therapy showed promising outcomes: 50% of patients of the first cohort showed no radiological progression over 6 months, and the other half of the patients were stable over 16 months. Three patients of the second cohort developed mild cytokine release syndrome that was self-resolving [111]. Based on this outcome, many clinical trials of anti-PSMA CART-T cells are underway [105,106]. P-PSMA-101 CAR-T cells are autologous purified CART-T cells developed through a novel technique. During manufacturing, the genes are inserted in a DHFR selection gene, anti-PSMA centyrin CAR gene and iCASP-9-based safety switch to purify CAR-T cells. They are characterized by a lot of memory stem T cells [112]. In 2022 GU ASCO data, P-PSMA-101 CAR-T cells demonstrated a significant antitumor effect in patients with mCRPC (clinical trial NCT04249947). Seven of ten heavily pre-treated patients with mCRPC achieved a PSA decrease in the preliminary report [112]. PSMA-targeting TGF-β-insensitive armored CAR-T cells are designed to overcome the suppressive immune microenvironment of PCa [113,114,115]. In phase I clinical trials (NCT03089203, NCT04227275), the preliminary results showed that TGF-β-resistant CAR-T cells are safe and feasible in patients with mCRPC [113,116], and further results are expected to unveil the drug distribution and disease response.

Prostate Stem Cell Antigen (PSCA) is highly expressed in PCa tissues. Although some pre-clinical trials showed that tumors can escape PSCA CAR-T cells [117], many clinical trials of anti-PSCA CAR-T cells, such as phase I study (NCT03873805) to examine the efficacy of PSCA-specific CAR-T cells in patients with PSCA-positive mCRPC [118] and clinical trial (NCT02744287) to study the safety and tolerable dose in patients with PSCA-positive CRPC or pancreatic cancer [119], are still in progress.

## 8. Targeted Therapies

### 8.1. Polyadenosine Diphosphate Ribose Polymerase (PARP) Inhibitors

PCa is known as a multiple-focal disease involving abnormalities in a variety of growth factor signaling pathways and aberrations in DNA damage repair pathways [120]. The analysis of germline mutations revealed that approximately 11.8% of mPCa patients harbor mutations in genes-related DNA-repair processes, including BRCA2, ATM, CHEK2 and BRCA1 [121]. Another study reported that an even higher percentage (27%) of PCa patients with local or metastatic disease harbor a germline or somatic alteration in the DNA repair gene [120]. Repair of strand breaks by homologous recombination (high-fidelity) and non-homologous (low fidelity) end-joining are two important mechanisms [122,123] that help cells repair from double-strand breaks, which are very hazardous DNA damage in cells.

PARP is a kind of enzyme that helps repair DNA damage in cells. It is essential for the repair of single-strand DNA breaks via the base excision repair pathway, which is a primary repair pathway for DNA single-strand breaks [124,125]. PARylation or poly-ADP ribosylation is the process by which PARP1 and PARP2 enzymes can sense single-strand DNA breaks and recruit DNA repair complexes to the site in the nucleus and result in post-translational modification of the DNA [124].

PARP inhibitors can block this PARylation, and they may also trap the PARP enzymes on injured DNA, preventing binding of incoming repair proteins [124]. The trapped PARP-DNA complexes are more cytotoxic than unrepaired single-strand breaks caused by PARP inactivation alone [126] (Figure 1).

### 8.2. Olaparib

The phase II trials demonstrated that germline and somatic mutations in homologous recombination repair (HRR) genes are predictive of the clinical benefit of PARP inhibition in PCa [127]. These genes include BRCA1/2, ATM, Fanconi’s anemia genes, and CHEK2 [127]. It showed that patients who had no response to standard treatment including enzalutamide, abiraterone or cabazitaxel and had DNA repair genes defect will get a high response rate to the PARP inhibitor (Olaparib) [127]. In a randomized, open-label (PROfound), phase III trial of olaparib demonstrated the benefit of disease progression-free survival rate in mCRPC patients who received a new hormonal agent (e.g., enzalutamide or abiraterone) and carried alterations in genes involved in HRR [30] (Table 1). The FDA approved the use of Olaparib in 2020 for patients with mCRPC with at least one germline or somatic mutation in HRR genes (BRCA1, BRCA2, ATM, BARD1, BRIP1, CDK12, CHEK1, CHEK2, FANCL, PALB2, RAD51B, RAD51C, RAD51D, or RAD54L) and who have previously received enzalutamide or abiraterone [93].

In a phase II study, olaparib in combination with abiraterone exhibited significant clinical efficacy benefits for patients with mCRPC compared with abiraterone alone [128]. Furthermore, this combination therapy clearly demonstrated a significant improvement in rPFS compared to abiraterone alone (24.8 vs. 16.6 months *p* < 0.0001) from a phase III clinical trial (PROpel study) that enrolled newly diagnosed mCRPC patients regardless of the HRR status [129]. Thus, this combination therapy is expected to provide potential therapeutic benefits for mCRPC patients in the future [129].

In addition, some studies indicated that mutant SPOP can deregulate DNA double-strand break (DSB) repair by promoting the error-prone non-homologous end-joining (NHEJ) pathway, like the breast cancer gene 1 (BRCA1) effect and increase the sensitivity of PCa cells to the PARP inhibitor [130]. The therapy outcomes are best for mutant SPOP patients without other (TP53, homologous recombination deficiency (HRD) pathway and PI3K/AKT pathway) concurrent mutations [43].

### 8.3. Rucaparib

Rucaparib is the second PARP inhibitor which the FDA approved for the use of mCRPC [93]. In a phase II trial (TRITON2), it evaluated 115 patients with a BRCA alteration with or without measurable disease and demonstrated primary end points with an objective response rate of 43.5% (95% CI, 31.0% to 56.7%; 27 of 62 patients). The most treatment-related adverse effect was anemia [31] (Table 1).

A phase III (TRITON3) study of PARP inhibitor rucaparib vs. the physician’s choice of therapy (abiraterone, enzalutamide, and docetaxel) for patients with mCRPC associated with homologous recombination deficiency (HRD) who have previously been treated with novel hormone agents and the results are pending [32] (Table 1).

### 8.4. Niraparib

Niraparib is a potent inhibitor of PARP-1 and PARP-2. In the phase III clinical trial (MAGNITUDE study), niraparib combined with abiraterone significantly improves rPFS, time to PSA progression and time to cytoloxic in patients with HRR-positive mCRPC [131]. For patients in the HRR-negative group, no clinical benefit has been observed. Thus, the study warrants the importance of genetic testing for achieving the therapeutic efficacy benefit of combination therapy.

### 8.5. Talazoparib

Talazoparib is a FDA-approved PARP inhibitor for BRCA-mutated, HER2-negative locally advanced or metastatic breast cancer. Currently, another phase I clinical trial (NCT 04703920) is ongoing to study the safety and tolerable dose of talazoparib in combination with belinostat for metastatic breast cancer, mCRPC, and metastatic ovarian cancer [132].

## 9. Phosphatidylinositol-3-Kinase (PI3K)/AKT Inhibitor

The PI3K/AKT/mTOR pathway is an important mutated pathway in PCa and tumor suppressor phosphate and tensin homolog (PTEN) is the most important negative regulator in this pathway [133,134,135]. Many studies indicated that loss of PTEN may be related to poor prognosis of PCa [133,134,136]. PTEN loss can activate the pathway of the PI3K/AKT/mTOR pathway and regulate the AR transcriptional output associated with hormonal resistance [137,138] (Figure 1). Loss of PTEN expression also demonstrated poor survival response in patients with CRPC to abiraterone [138] and resistance to PARP inhibitor [75]. Therefore, a combination with the PI3K/AKT pathway inhibitor and the PARP-inhibitor is currently a new way to overcome the resistance [139]. In the phase III clinical trial (IPATential150), the ipatasertib (AKT inhibitor) plus abiraterone group significantly improved the rPFS of the patients with mCRPC with PTEN loss compared to the placebo plus abiraterone group (18.5 vs. 16.5 months *p* = 0.034) [140]. Although the combination therapy failed to show significant improvement of rPFS in the intention-to-treat population, this trial does support the role of PTEN status in determining targeted therapeutic strategy.

## 10. Aurora Kinase Inhibitor

### Alisertib

Beltran et al. indicated that the oncogenic transcriptional factor N-Myc and cell cycle kinase Aurora kinase A (AURKA) are overexpressed in most metastatic neuroendocrine PCa (mNEPC), a subset of CRPC [141] (Figure 1). This study demonstrated that alteration of N-Myc and AURKA are involved in the development of NEPC and Aurora kinase inhibitor shows benefits for NEPC. The AURKA inhibitor, alisertib, disrupts the interaction between the N-Myc-Aurora A protein complex and inhibits N-Myc signaling tumor growth [142]. In a phase I study (NCT01094288), alisertib combined with docetaxel demonstrated antitumor activity. Six of the eighteen patients with CRPC had partial responses [143]. A single-arm, multi-institutional open-label phase II clinical trial (NCT01799278) showed that a subset of patients with advanced PCa (CRPC or NEPC) with molecular features in favor of Aurora-A and N-myc activation achieved significant benefits from a single agent with alisertib [144]. The inclusion criteria of the trial contained patients with metastatic PCa and at least (1) small-cell neuroendocrine morphology, (2) more than 50% neuroendocrine marker expression, (3) development of liver metastasis without PSA progression, or (4) elevated serum neuroendocrine marker [144]. In the molecular analysis, AURKA amplification was associated with OS improvement (*p* = 0.05) in alisertib but no difference in PFS (*p* = 0.4). Although N-myc amplification was not associated with outcomes, the responders still showed correlation with MYCN overactivity [144]. However, in another phase I/II study (NCT01848067), combination therapy with alisertib did not reveal significant benefits in patients with CRPC progressing on abiraterone [145].

## 11. MEK Inhibitor

### Trametinib

NEPC is associated with being highly aggressive and associated with EMT and neuroendocrine transdifferentiation (NEtD) [146,147]. Clinically, EMT induction and mesenchymal characteristics are associated with high Gleason grade, biological recurrence and visceral metastasis [8]. NEtD is an important route and the development of targeted therapeutics, which can reverse the molecular and phenotypes traits, may be a possible therapy. Cells transdifferentiate into the neuroendocrine phenotypes such as low AR activity, high cellular plasticity drivers [(Axl, mitogen-activated protein kinase (MEK), AURKA, Brachyury)], high-grade, and low PSA [146] (Figure 1). A phase II clinical trial (NCT02881242) of trametinib, a MEK1/2 inhibitor, is currently ongoing for patients with refractory metastatic PCa [148].

## 12. Histone Deacetylase Inhibitors (HDAC Inhibitors)

Epigenetic aberrations are known to cause cancer development [149,150,151]. Histone protein balance is controlled by histone acetyl transferases (HATs) and histone deacetylases (HDACs) [152,153] (Figure 1). The change of the histone protein balance influences many genes including oncogenes, tumor suppressor genes, and DNA repair genes [152,153]. Therefore, abnormal HDACs expression is related to the development of cancer [154]. Overexpression of HDAC1, 2 and 3 has been reported in PCa [155,156], and many HDAC inhibitors have been designed for PCa [157,158]. HDAC inhibitors demonstrated promising results in hematological malignancies and the FDA-approved suberoylanilide hydroxamic acid (SAHA, vorinostat) for the treatment of cutaneous T-cell lymphoma in 2006 [159]. However, four HDAC inhibitors (vorinostat, pracinostat, panobinostat and romidepsin) failed in phase II clinical trials for CRPC because of drug toxicity and disease progression [160]. Nevertheless, further investigation for more specific HDAC inhibitors targeting HDAC subtypes still can be used as a promising PCa therapy.

## 13. SPHK1/S1PRs Inhibitors

It has been found that sphingolipid metabolites are involved in the development of many cancers. Sphingosine is phosphorylated by SPHK1 (*Sphingosine kinase* type 1) and then becomes sphingosine-1-phosphate (S1P). The S1P is able to bind on the S1P receptors 1, 2, 3, 4, and 5 (S1PRs) for further signaling transduction, including ERK1/2, PI3K/AKT, NF-κB, STATs, Rho/ROCK, and RAS [161]. It also has been found that SPHK1/S1PRs signaling promotes cancer cell motility, angiogenesis, survival, and proliferation [162]. Currently, many cancer studies demonstrate that SPHK1/S1PRs inhibitors, such as MP-A08, SKI-I, SKI-II, FTY720, LCL146, LCL351, SLP7111228 and B5354c, are able to suppress cancer cell proliferation or apoptosis induction in breast cancer, bladder cancer, leukemia cells, and PCa [163,164,165]. Also, recent studies demonstrate that SPHK1/S1PRs-targeting treatment can have a significant impact on PCa cell growth [166], angiogenesis [167], chemotherapy resistance [164,165,168,169,170], radiotherapy resistance [171], migration and invasion [172]. Notably, elevated SPHK1 activity is detected in PCa specimens as compared to normal tissues, suggesting the clinical prevalence of SPHK1/S1PRs in PCa development [173].

Our recent study indicates the critical role of the SPHK1/S1PRs pathway in PCa cell migration and invasion via matriptase regulation [172]. Furthermore, the SPHK1/S1PRs pathway is involved in neuroendocrine PCa development through REST degradation [174]. In these studies, fingolimod (FTY720), an FDA-approved SPHK1 inhibitor (Figure 1) for treating multiple sclerosis [175], shows significant therapeutic efficacy in clinically relevant NEPC xenograft models, supporting the future clinical translation of targeted therapeutics for NEPC.

## 14. PTP1B Inhibitors

In recent years, the protein tyrosine phosphatase 1B (PTP1B; also known as PTPN1) has emerged as a critical regulator of multiple signaling networks involved in human disorders, such as obesity [176], diabetes [177], and cancers [178]. Moreover, several studies point toward PTP1B serving as a potential therapeutic target in various tumors, such as PCa [179], pancreatic cancer [180], ovarian cancer [181], colon cancer [182] and breast cancer [183]. Wu et al. indicated that PTP1B elevation was detected in neuroendocrine differentiation in PCa specimens [184]. Also, one study suggested that PTP1B deletion or inhibition (PTP1B inhibitor; MSI-1436) could enhance T-cell antitumor activity and improve the therapeutic efficacy of chimeric antigen receptor (CAR) T cells in solid tumors [185] (Figure 1). Generally, the accumulative evidence suggested that PTP1B may serve as a promising therapeutic target for t-CRPC treatment.

## 15. Conclusions

Despite the fact that the introduction of several advanced therapeutic innovations has shown to improve survival and reduce morbidity and mortality of mCRPC, the onset of t-CRPC imposes a new challenging for curing PCa. Based on the molecular characterizations of subpopulations of t-CRPC, a variety of new agents such as genetically targeted agents, specific small-molecule inhibitors, and immune therapies are underway to explore the precision medicine treatment for individual PCa patients.

## Figures and Tables

**Figure 1 biomedicines-10-01872-f001:**
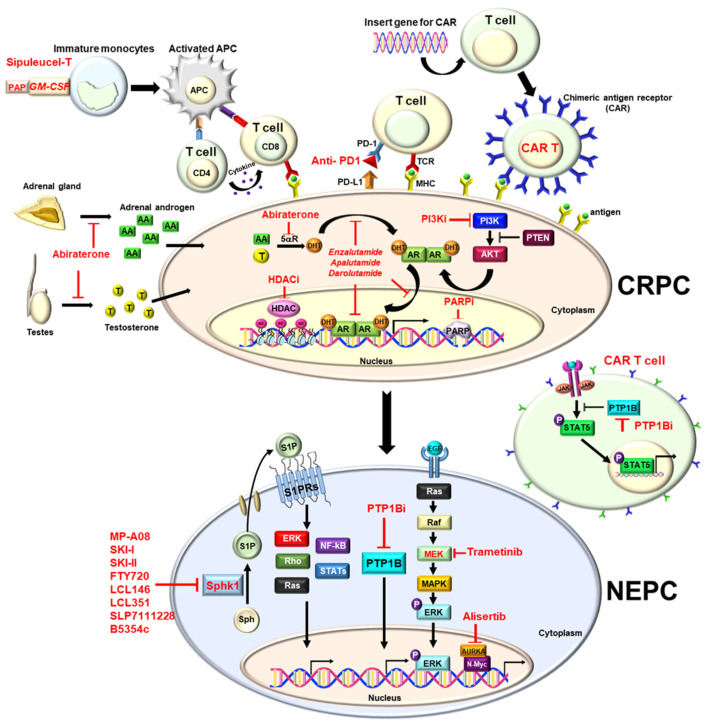
Molecular mechanism of the therapies for prostate cancer.

**Table 1 biomedicines-10-01872-t001:** FDA-approved therapies for prostate cancer.

Drug Class	Therapy	Indication	Clinical Trial	Efficacy (Mon.)	Impact on PCa Treatment	Reference
**2nd-generation antiandrogen**	Enzalutamide	mCRPC	AFFIRM	OS 18.4 vs. 13.6	ENZ improves OS after chemotherapy	[14]
			PREVAIL	OS 35.3 vs. 31.3	ENZ improves OS in chemotherapy naïve	[15,16]
		nmCRPC	PROSPER	MFS 36.6 vs. 14.7	ENZ decreases the risk of metastasis	[17]
		mCSPC	ENZEMET	80% vs. 72% (3-year OS)	ENZ improves OS and decreases disease progression	[18]
			ARCHES	rPFS NR vs. 19.0	ENZ decreases the risk of metastatic progression	[19]
	Apalutamide	nmCRPC	SPARTAN	MFS 40.5 vs. 16.2	APA decreasse the risk of metastasis	[20]
		mCSPC	TITAN	2-year OS 82.4 vs. 73.5	APA improves OS and PFS	[21]
	Darolutamide	nmCRPC	ARAMIS	MFS 40.4 vs. 18.4	DA decreases the risk of metastasis	[22]
	Abiraterone	mCRPC	COA-301	OS 15.8 vs. 11.2	AB improves OS after chemotherapy	[23]
			COA-302	OS 34.7 vs. 30.3	AB improves OS in chemotherapy naïve	[24]
		High-volume CSPC	LATITUDE	OS 53.3 vs. 36.5	AB improves OS in high-risk CSPC	[25,26]
**Radiotherapy**	Radium 223	CRPC with symptomatic bone metastasis, no visceral metastasis	ALSYMPCA	OS 14.9 vs. 11.3	RA-223 improves OS in symptomatic bony metastatic mCRPC	[27]
	^177^Lu-PSMA 617	PSMA-positive mCRPC and already treated with ARB and chemotherapy	VISION	OS 15.3 vs. 11.3 rPFS 8.7 vs. 3.4	^177^Lu-PSMA 617 improves OS and rPFS in PSMA-positive mCRPC	[28]
**Immunotherapy**	Sipuleucel-T	mCRPC	IMPACT	OS 25.8 vs. 21.7	Sipuleucel-T improves OS in mCRPC	[29]
**PARP-I**	Olaparib	mCRPC with HRR genes mutation after ENZ or AB	PROfound	OS 18.5 vs.15.1 rPFS 7.4 cs. 3.6	Olaparib improves OS and rPFS in mCRPC with HRR gene mutation	[30]
	Rubraca	mCRPC with BRCA genes after ARB or chemotherapy	TRITON2 (phase2) TRITON3 (phase3)	ORR 43.5% (IRR) PSA response rate 54.8%	Rubraca has the antitumor activity in mCRPC with BRCA gene mutation	[31,32]

PCa = prostate cancer; nmCRPC = non-metastatic castration-resistant prostate cancer; mCSPC = metastatic castration-sensitive prostate cancer; mCRPC = metastatic castration-resistant prostate cancer; OS = overall survival; rPFS = Radiographic progression-free survival; FDA = Food and Drug Administration; ENZ = Enzalutamide; APA = Apalutamide; DA = Darolutamide; AB = Abiraterone; MFS = metastatic-free survival; NR = non-reached; RA-223 = Radium 223; ARB = androgen receptor blocker; HRR = homologous recombination repair; ORR = objective response rate; IRR = independent radiology review; PARP-I = Polyadenosine diphosphate ribose polymerase inhibitors.

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
