# Peer review of "Developing New Treatment Options for Castration-Resistant Prostate Cancer and Recurrent Disease"

_biomedicines, 2022, doi:10.3390/biomedicines10081872_

Round 1

Reviewer 1 Report

Overall , this is a well-written review on this topic. The authors did a comprehensive review on FDA-approved treatment options. However, in the perspective of “Developing new treatment options”, newer treatment options that are currently in different clinical trials phases are not well covered. Below are some major and minor points.

1.       In the hormone therapy section, some novel androgen receptor inhibitors options currently in clinical trials were not discussed, such as the chemical degradation of AR using PROTACs. It at least include ARV-110, ARV-471, ARV-766, AR-LDD, ARCC-4, ARD-61 in clinical trials

2.       In CAR-T therapy section, some phase I clinical trials were missed:

·         P-PSMA-101 CAR-T Cell, which are enriched for memory stem T cells.

·         CART against prostate stem cell antigen (NCT03873805 and NCT02744287)

·         Additionally, different CAR-T strategies were not discussed. To name a few: auto vs allo, TGF-b armored vs enrichment CAR-T, CAR-T in combination with lymphodepletion,

3.       In PARPi section, it is worth mentioning other inhibitors that are currently in clinical development, such as niraparib, talozoparib, and veliparib.

Minor:

The last sentence in line 51-52 is confusing

Confusing sentence in line 133-134

Author Response

  1. Yes, thanks for your constructive comment. A new paragraph (Page 7) was added to correspond this comment.  
  2. Yes, thanks for your suggestion. We have added more descriptions (see Page 12-14) that cover P-PSMA-101 CAR-T cells, clinical trials of

    PSCA-specific CAR-T cells (NCT03873805 and NCT02744287) and PSMA-targeting TGF-β insensitive armored CAR-T cells. Also, we compare the different strategies including autologous and allogeneic CAR-T cells, CAR-T in combination with lymphodepletion.

  3.  Thanks for your suggestion; two corresponding paragraphs for niraparib, talozoparib were added (see Page 16).

    Minor points:

    Thanks for your comment. We have modified accordingly (see Page 3, 2nd paragraph, Line 4-10).

    Thanks for your comment. We have modified accordingly (see Page 6, 1st paragraph, Line 14-18).

Reviewer 2 Report

The manuscript present an interesting review of the approach into the treatment of prostate cancer. It is important for this population to have more alternatives into treating this cancer, that develop metastatic diagnosis and put the lives of the patients in danger. It will be interesting to keep an eye of the development of new treatments that could potentially help these patients into a complete remission state. 

There are some corrections that needs to be made: 

-FDA stands for Food and Drug Administration not Food and drug association. 

Author Response

Yes, thanks for your correction. We have changed it accordingly and asked editor to improve writing.

Reviewer 3 Report

In this investigation Wang et al. explained the new therapeutic options for the treatment of castration resistant prostate cancer (PCa) and its clinical outcomes. The topics covered in this review were self-explanatory and extensive covered vast corners of relevant research. New treatment options is much needed therapeutic entity for prostate cancer treatment, with much of clinical impact. However, Authors need to address the following concerns before acceptance.

1)    Schematic illustrations pertaining to their molecular actions of the described drugs should be included.

2)    The text pertaining shortcomings related to the existing therapies in PCa. And How this new therapeutic window will offer the better outcomes. Explain the strategic approaches to be taken for overcoming this issue.

3)    Topics related inflammatory responses towards various new therapeutics in PCa patients were missing and can be covered.

4)    Insert the table with the list of FDA approved therapies against PCa. Also cover the clinical success of those drugs. It will ease the readers to understand the impact of ADCs in cancer therapy.

5)    The font size of text in some of the areas all through the manuscript was not uniform and can be rectified.

6)    The manuscript can be further revised for grammatical and typological errors.

Author Response

  1. Yes, thanks for your suggestion. We have added an illustration to indicate the molecular mechanism of each agent.
  2. Yes, we have mentioned several strategies approaches to overcome the shortcomings of the current therapies.

  • Combination therapies 

           (a) PTEN loss: Ipatasertib (AKT inhibitor) plus abiraterone group significantly improve the rPFS of the patients with mCRPC with PTEN loss (see Page 16, 4th paragraph, Line 8-12)

             (b) Phase III clinical trial (PROpel study): It demonstrated significant improvement in rPFS for Olaparib + Abiraterone vs placebo + Abiraterone (HR 0.66, 95% CI 0.54-0.81) in patients with newly detected mCRPC, irrespective of HRR status (see Page 15, 2nd paragraph).

  • Immunotherapy: Immunotherapy usually failed in PCa because of immunosuppressive environment and many CAR-T therapy, like P-PSMA-101 CAR-T cells, PSMA-targeting TGF-β insensitive armored CAR-T cells are developed to overcome the immunosuppressive environment (see Page 13).
  • Precision medicine : 177Lu-PSMA 617 improve OS and rPFS in PSMA-positive mCRPC (see Page 9-10).

3.  Yes, thanks for your suggestion. We have added discussion (see Page 6, 1st paragraph, Line 18-22) for this topic.

4. Yes, thanks for your suggestion. A table containing FDA approved regimens was added.

5. Thank you for your suggestion. We have made all the corrections.

6. Thank you for your recommendation. We have asked an English professional to edit this manuscript.

Round 2

Reviewer 3 Report

Satisfied with the Author's response.